# Climate History of the Principality of Transylvania during the Maunder Minimum (MM) Years (1645–1715 CE) Reconstructed from German Language Sources

**Martin Stangl [1,\*] and Ulrich Foelsche [1,2]**

1   Institute for Geophysics, Astrophysics, and Meteorology/Institute of Physics (IGAM/IP), University of Graz, 8010 Graz, Austria; ulrich.foelsche@uni-graz.at
2   Wegener Center for Climate and Global Change (WEGC), University of Graz, 8010 Graz, Austria
\*   Correspondence: martin.stangl@uni-graz.at

**Abstract:** This paper deals with the climate in the former Grand Duchy of Transylvania, now one of the three major geographical provinces of Romania, within the so-called Maunder Minimum (MM) (1645–1715), an astrophysically defined part of the Little Ice Age (LIA), which was characterized by reduced solar activity. The historical data from Transylvania are compared with that from Germany, Austria and Switzerland. This comparison for the period 1645–1715 shows good agreement but also reveals geographic characteristics of the region. For the first time, we present here a comparison between the four geographic areas in text and tabular form. Quotes from mostly German-language sources are reproduced in English translation. The results clearly help to identify regional climatic differences during the MM. Furthermore, we examine for a longer period (1500–1950) the extent to which the climate of Transylvania might have been affected by long-term fluctuations in solar activity, as deduced from isotopic reconstructions from ice cores. This way we compared astrophysical conditions with climatological ones in order to see if any probable relations do indeed show up. This comparison suggests a certain solar influence but the agreement is not very pronounced. Future investigation in a pan-European context is needed to reach reliable statements. Some results are unexpected—such as an unusually small number of severe winters during the last decades of the MM, where extreme cold was restricted to a few years, like the extreme winters 1699/1700 and 1708/1709.

**Keywords:** climate of Middle Europe; Maunder Minimum; Little Ice Age; solar activity

## 1. Introduction

The period of 1645–1715 is called Maunder Minimum (MM) in solar physics and denotes an epoch of long-term reduced solar activity within the long-known eleven-year activity cycle. The modern discoverer of this seventy-year minimum period was the American astrophysicist John Eddy. He caused a sensation in a paper in *Nature* in 1976 [1], especially because he postulated a causal connection with the Little Ice Age (LIA). Since then, the term MM has also been used by climatologists, but not always in its correct meaning. The MM is to be understood here strictly as the astrophysical definition of the calendar years from 1645 to 1715.

From the very beginning, when it was recognized, that our climate is not constant, climatic variations were attempted assumed to be related to the sun [2], but the more precisely it became possible to measure solar irradiance, the more it became apparent, that the fluctuations of the solar constant registered so far are too small to directly justify climatic changes. However, it is generally supposed that the sun affects the climate indirectly. Usoskin and Kovaltsov [3] suggested a possible process of ionization by cosmic rays, while Ney [4], half a century before, suspected an increasing and decreasing frequency of cloudiness due to minor temperature differences.

Over the solar cycle, the irradiance changes mainly in the UV part of the spectrum, which could lead to a weakening and widening of the tropical Hadley cells and to shifts of the subtropical jet streams, or to a shift of the intertropical convergence zone [5]. Theoretical considerations, however, assume that even in the case of a grand, i.e., long duration minimum, as it is assumed for the MM, no significantly reduced irradiance should have occurred [6]. Therefore, indirect mechanisms, such as changes in air currents and pressure patterns in the troposphere due to different heating of the stratosphere, seem more plausible [7]. In any case, other factors, such as volcanic eruptions or the El Niño phenomenon, also seem to have a much more drastic effect on cloud formation than fluctuations in solar irradiance.

Lockwood et al. [8] supposed that low solar activity leads to cold winters on the British Isles and parts of continental Europe, while leaving most of continental Europe untouched. According to this study, European climate, lying below affected by the polar jet stream, might be influenced by solar activity, in such a way that it "blocks" the jet stream and thus upholds the mild, humid westerly winds that otherwise characterize English winter. Such "blockings" might push them further north, replacing them with cold, dry easterly winds over parts of Europe. However, it is open to discussion to what extent fluctuations in solar activity change stratospheric winds, given the small magnitude of the changes in irradiance between solar maximum and minimum, and presumably also during the MM.

Mounting evidence shows, that the climate of the MM—and the LIA in general—was neither globally synchronous nor regionally coherent [9], therefore it is necessary to study historical records (and proxy data) from different regions. While several studies from western and central continental Europe, as well as from the British Isles, have been carried out on the climate of the grand minimum period [10], the Eastern parts of Central Europe have so far been rather neglected in comparative studies of solar forcing. We try to contribute to closing this gap [11] by using historical sources from Transylvania. Results on historical records of solar activity from the same region have just recently been published in Solar Physics [12].

Extensive work in order to look Europe-wide into the problem has been done within the frame of the EU project ADVICE. Refs. [13,14] and important conclusions have been reached regarding the diversity of the climate of the MM over the different parts of Europe. While it is difficult to compare these results, which include instrumental data as well as sophisticated reconstructions of sea level pressures, with our findings, general patterns regarding the individual years can help to fill in the gaps in a region with sparse data, as it is the case with Transylvania. While quantitative temperature and precipitation reconstructions from the region have already been used in the 500-year Central European temperature reconstruction of Dobrovolný et al. [15], it was not our purpose here to dilute local data by incorporating them into a larger geographical context and therefore lower their spatial resolution, but rather work out the differences between specific regions, in our case different regions of German speaking settlement areas.

In the present paper, we look at the situation in the former Grand Duchy of Transylvania, which was initially dependent from the Ottoman Empire and then (since 1699) incorporated into the Habsburg Empire. Today, Transylvania is only a geographical term. It describes one of the three major parts of the Republic of Romania, which consists of the major regions of Wallachia, Moldova and Transylvania. Individual authors have already dealt with issues of climate history of that region in the past [16–19], but a study looking into possible solar forcing to understand those data has so far been lacking. To achieve this, we consulted original sources of the time and were therefore able to reconstruct the MM climatically for Transylvania on an annual basis. Finally, we summarize historical records for the individual years, most often in the form of English translations from German originals, where we try to conserve the original ductus. Further the material from Transylvania is compared with that from Germany, Austria and Switzerland. Finally, we investigate for a more extended period (1500–1950) whether the climatic conditions of the time in Transylvania actually indicate solar forcing or not.

## 2. Materials and Methods

Transylvania is one of the three major geographical regions that constitute today's state of Romania and is of comparable geographical latitude with Austria and Switzerland, although the Carpathian Mountains do not reach the heights of the Alps. Today's climate of Transylvania can predominantly be characterized with Dfb (warm summer continental or hemiboreal climate) and Cfb (oceanic climate) according to the Köppen scheme [20].

All cities and villages mentioned in the text are given in Romanian language, but are listed in Table 1 in the three major languages spoken within the region, i.e., in Romanian, Hungarian and German. Mountains, rivers and other landmarks appear in their official designations in Romanian, but Hungarian or German names are given in many instances in brackets to ease their identification in historical sources.

**Table 1.** List of localities mentioned in the text, given with their names in the three major spoken languages of the country and their approximate coordinates and altitudes.

| Romanian | Hungarian | German | Approx. Coordinates | Approx. Altitude |
|---|---|---|---|---|
| Biertan | Berethalom | Birthälm | 46°08′ N/24°31′ E | 388 m |
| Bod | Botfalu | Brenndorf | 45°46′ N/25°39′ E | 506 m |
| Brașov | Brassó | Kronstadt | 45°40′ N/25°37′ E | 600 m |
| Cluj-Napoca | Kolozsvár | Klausenburg | 46°46′ N/23°35′ E | 340 m |
| Codlea | Feketehalom | Zeiden | 45°41′ N/25°26′ E | 565 m |
| Hălchiu | Höltövény | Heldsdorf | 45°46′ N/25°33′ E | 507 m |
| Hărman | Szászhermány | Honigberg | 45°43′ N/25°41′ E | 529 m |
| Mediaș | Medgyes | Mediasch | 46°10′ N/24°21′ E | 330 m |
| Năsăud | Naszód | Nösen | 47°17′ N/24°24′ E | 331 m |
| Râșnov | Barcarozsnyó | Rosenau | 45°36′ N/25°28′ E | 635 m |
| Sebeș | Szászsebes | Mühlbach | 45°58′ N/23°34′ E | 248 m |
| Sibiu | Nagyszeben | Hermannstadt | 45°48′ N/24°09′ E | 431 m |
| Sighișoara | Segesvár | Schäßburg | 46°13′ N/24°47′ E | 380 m |
| Vulcan | Szászvolkány | Wolkendorf | 45°38′ N/25°25′ E | 606 m |
| Zărnești | Zernest | Zernescht | 45°34′ N/25°20′ E | 716 m |

The richest sources regarding our study period, i.e., the 17th and 18th centuries, can be found in the southern and eastern parts of Transylvania [21]. Extensive chronicles like those written by Miles [22] or Kraus [23] complement casual notes with relevance for climate history that have been found, among them in the so-called Turmknopfschrift ("poppyhead document") deposited in August 1794 in the poppy head of the Protestant church of Zeiden (today Codlea) written by the former pastor of this place and afterwards recovered and published about a hundred years later (Draudt 1903). While Matthias Miles (1639–1686) describes events from the century preceding his lifetime, he is regarded a very reliable source by historians [21,24]. An eyewitness of the events, or at least a contemporary observer for the events he was writing about, was the city clerk of Sighisoara, Georg(ius) Kraus(s) (1607–1679). In 1650, he started with working on his Siebenbürgische Chronik (Transylvanian Chronicle), which deals with local events between the years 1608 and 1665, although the events between 1650 and 1665 get the most intense treatment and occupy about 75% of the whole chronicle. While Kraus is regarded as the most important chronicler of his time, for our purpose of extracting clues with climatic relevance, the clumsy written chronicle of his contemporary Daniel Nekesch Schuller (born in 1606—year of death unknown) is a somewhat richer source. An important source for our study have

been diaries, most notably those by Johannes Stamm, a linen weaver and Paul Benckner, a trader, both from Brașov.

The results from Transylvania are confronted with data from Austria, Germany and Switzerland (understood in their current borders). The climatological history of Germany and Switzerland has been extensively studied in monographs by Glaser [25] and Pfister [26], while for Austria only local compendia exist and for more limited time spans, most notably the studies of Jäger [27] for the Tyrol region and Strömmer [28] for the region of Lower Austria. However, extensive older literature and sources exist and many years in our study period are well documented by Pilgram [29] and Peinlich [30,31]. As for Transylvania, the by far most extensive study in English language is the one by Rácz [18], while further data were collected by Réthly [16], but are available only in Hungarian. For our study we also consulted Cernovodeanu and Binder [19], which is available in the Romanian language only. Figure 1 shows the title page of a typical contemporary source, the "Siebenbürgischer Würg-Engel" by Matthias Miles.

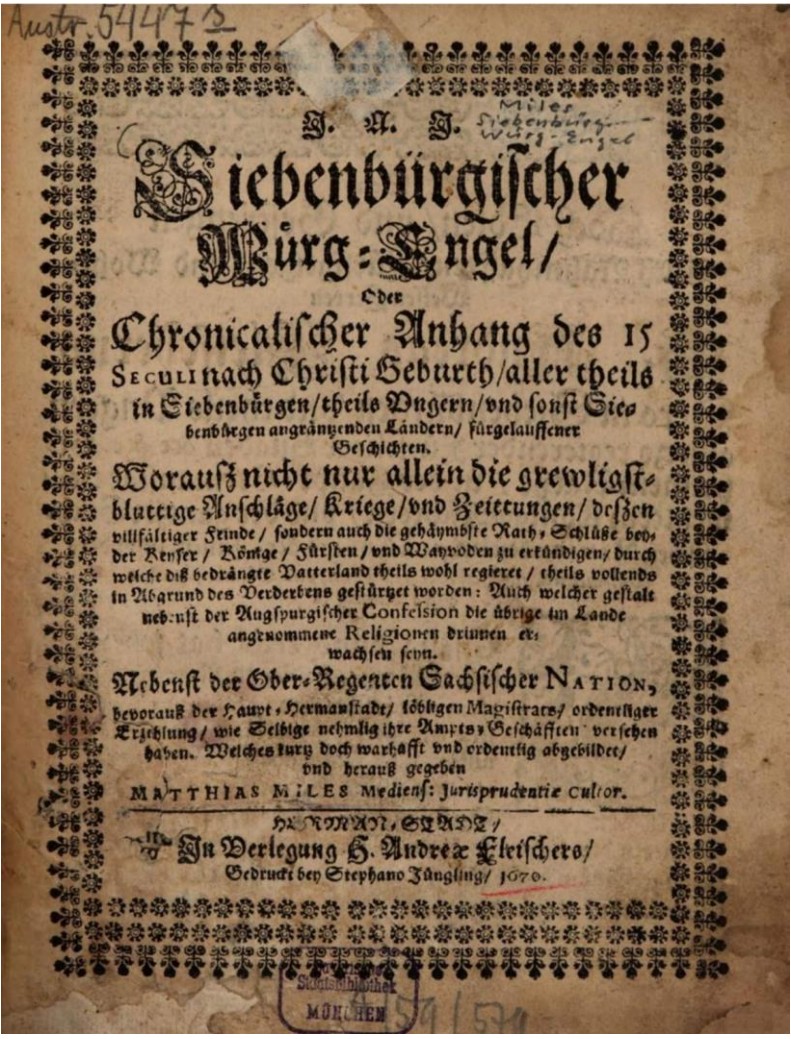

**Figure 1.** Title page of Matthias Miles' Siebenbürgischer Würg-Engel, which reads "Transylvanian destroying angel or chronological appendix of the stories that occurred in Transylvania, partly in Hungary and in the other countries bordering on Transylvania in the 15th century AD, from which not only the horrific bloody attacks, wars, and news from various enemies of the country emerge, but also the most secret decisions of all emperors, kings, princes, and voyvods, through which its oppressed fatherland flourished in part, in part completely plunged into the abyss of ruin: in what manner besides the Augsburg confession the rest of the country's accepted religions have emerged". Courtesy Bayerische Staatsbibliothek München.

### 3. Results

*3.1. Annual Characteristics of the MM in Transylvania*

The year 1645, the first year that solar physicists attribute to the MM, began with an extremely cold winter [32] (p. 40). The summer brought a plague of locusts [33] (p. 394) and severe weather conditions: "On the day of St. John the Baptist (24 June) a large flood of water poured out and caused great damage. On the 11th day in July, the day of Eleneora, there was such a big hail here in Cronen (Brașov) that one thought everything would get destroyed, the fruits of the earth destroyed in the whole field; and then, after a short time there was such a big wind that one thought that everything would go to ruin; great damage has occurred outside in the old town, in the barns and on the roofs." [34] (p. 229). The winter of 1646 did not seem unusually harsh to people, but on 29 March, the cold returned for a short time: "On Maundy Thursday, a substantial amount of snow fell." [34] (p. 230). The summer was very different from that of the previous year. According to Georg Kraus, the city clerk of Sighișoara, there was not a single thunderstorm throughout the summer [23] (p. 165).

1647 was notable for inundations [19] (p. 123), 1648 by a harsh winter [19] (p.123). In 1649, the winter was very long and severe. From December 6 of the previous year until April, the landscape was continually snow-covered and people, as well as livestock, suffered from the great cold without interruption: "Anno 1649 is starting with a very cold and strong winter, because snow, which fell on December 6 the previous year, remained until April, that is, for 4 months, during which the good grain was completely spoiled." [23] (p. 136). The autumn was characterized by long-lasting rainfall: "Around November, as well as in the previous month, because of the constant rain and pouring water, the soil is so soaked that everything, so to speak, has been swimming in the water; everywhere in Transylvania the roads have become bad, and no one could travel without tying up many additional good horses and oxen, especially through villages and swampy places, such as on the way to Nösen (Năsăud) ( . . . ); often, over 30 horses and oxen had to pull; (all of these) were signs of ruin, just as many mountains and hills were broken and devasted in many places, according to the Psalm that the Earth is tired of carrying such brood of hell (even longer); that is a sign of the Day of Judgement." [23] (p. 136).

1650 was apparently an unremarkable year. Only a strong storm in late winter seemed to be worth to report for the citizens of Brasov: "On 10 March, there was a terrible great wind at night that one thought everything was about to perish; (It) made a lot of damage to the barns and buildings of the old town and trees were torn down and it also caused great damage to the roofs in the city." [34] (p. 233). In 1651, the Mureș and other rivers caused inundations in January. A late frost at the end of April damaged the fruits: "On 26 April, heavy snow fell, which caused (such) a great and violent cold that the vineyards and cherry orchards were completely ruined by frost, as was not often seen at the time, and the cost of a bucket of grain was fl.[orin] 4.50 and fl.[orin] 3.30 at that time." [34] (p. 233). Strange weather conditions were noted in Sighișoara: "Anno 1651. On 3 May, it rained blood here in Segesvár (Sighișoara). In the current year, the time of ripeness came very late, and the cherries and strawberries ripened only in August, and the roses bloomed in September." [23] (p. 183). In 1652, the winter was severe. Swarms of locusts bared the land in Moldova and then covered the ground in winter, partly alive, partly dead. The plague was rampant in the country [35] (p. 487). In the town of Mediaș a thunderstorm in winter was interpreted as an omen by contemporaries: "Anno 1652. On 16 February, in cold winter weather in Mediaș, very great wind and heavy weather with thunder and lightning hit the Church tower. This and the subsequent weather are once again signs of the ruin of Transylvania." [23] (p. 188). For 1653, climate-relevant information from the country is missing. It can be assumed that the weather conditions met expectations, particularly that of farmers. In 1654, winter conditions prevailed for more than half a year, namely until April, and beginning with September, the next winter started. "This year has been such a long, violent winter that in Holy Week (at the beginning of April) one could ride a sleigh." [34] (p. 238). "Nota. When the fruits in the country of Nösen (Năsăud) at the foot of the mountains ripened very late, a lot of snow falls in September and covers everything,

so that the fruits have to be searched under the snow and it spoils a lot." [23] (p. 225). The short summer was characterized by floods: on 12 July, the sun was grass-green when it rose; this was followed by cloudbursts, which caused the streams and rivers to overflow their banks [36] (p. 166); on 23 August there was another flood [23] (p. 175).

The severe winter that started early did not subside even after the turn of the year to 1655 [19] (p. 123). "19 April brought a lot of snow due to God's punishment, so that in the gardens big fruit trees, grown 20 years ago, were torn from the earth with roots by the hundreds." [34] (p. 239). Additionally in 1656, the winter is noted as severe [19] (p. 123), and the cold of the following winter of 1657 was also harsh, albeit with interruptions. In the middle of winter, the army of Prince George II Rákóczi could hardly cross the river Tisza. Due to a great inundation, the waters were swelled due to sudden snowmelt in the mountains of the Maramures, but then the cold fell again, and the Tisza froze [37] (p. 298). In 1658 [19] (p. 123), and in the following years, there were many complaints about bitter winter cold. "In October (1658) great snow fell, and did so much damage to the trees in the gardens and in the field, (and) to the fruit. One evil followed the other. Nulla calamitas sola." [34] (p. 239). In 1659, the winter was hard and long [19] (p. 123). "So, on 31 May in the night of Pentecost Sunday, it was snowing (so much) that in the morning the trees could hardly be seen; (the snow) melted the next day." [38] (p. 177).

In 1660 the winter was also very cold, but this time it was dry. There was a drought in summer, that was interrupted by sudden cloudbursts [19] (p. 90). In 1661, the winter was again severe and lacked snow. The summer was dry too, but the drought was repeatedly interrupted by heavy cloudbursts [19] (p. 91). In 1662, the weather was largely similar to that of the previous year [19] (p. 123). "An unusual snow fell on 10 October, so you had to sow the grain in the snow." [34] (p. 263). Additionally in 1663, people complained about a very cold winter and the drought of summer, which was very harmful to the harvest [39] (p. II. 528). "March begins, which should drive all evil vapors out of the earth and dry them out by the cleaning wind, but many harmful veils of mist are now emerging." [34] (p. 277). In 1664, winter was no less severe. For the fifth time in a row, there was a dry year, interrupted only by excessive downpours in summer, which also harmed the harvest. "On 14 May, a very cold weather comes into (the time of) the flowering of the fruit, and there is also quite a bit of snow for two days; because we oppose God with disobedience, God also sweeps us out of his annual order with the summer becoming winter, whereby the blessings of God are withdrawn from all fruit; but, O Lord, do not go to judgment with your servants, but be kind and gracious to us and promote the work of our hands . . . Bears, big and small, 21, come into the old town and the gardens, do great damage and also hurt a lot of people; and it is to be feared that these bears will become human bears and pull the skin over our ears with the time . . . On 19 October, a heavy rainy weather mixed with snow develops for almost 8 days." [34] (p. 290).

In 1665, the winter was very cold again—"There was a lot of snow in all of Burzenland (Țara Bârsei), which did a lot of damage to the fruits." [40] (p. 112). The summer was plagued by heat and drought [19] (p. 123). The year 1666 has been described as a very fertile year [32] (p. 46). "It was also a difficult winter this year because the first snow that fell 14 days before Christmas Day (last year) remains very firm and heavy and persistent until March. Such an example (already) happened Anno 1608, and then there is a short and humid summer. Mense Aprili. At this time of spring, there comes a great dryness after the long winter." [41] (p. 16). 1667 has gone down in history as a rainy year in Transylvania [19] (p. 123). "(On) 22 July, the weather kills 3 Szeklers (people from Székely district, Transylvania) in the field in front of Brasov." [42] (p. 107). An extremely early onset of winter in Brasov, at the foot of the Carpathian Mountains, shocked the people who got snow one day in late summer and their fruit trees suffered breakage; some trees are said to have been knocked down completely: "Anno 1667. Nota bene. On 12 September, before dawn, there was so much snow, which fell at night, that many trees with fruit in the gardens were broken, and their roots were torn out." [38] (p. 191). 1668 was also a rainy year, and the warm part of the year was significantly shorter than the cold one. There was a

major flood in Sighișoara at the end of January; the water came into the houses, and people had to flee to the attics [32] (p. 46). Between 24 July and 5 August, there was constant rainy weather. The Târnava Mare overflowed its banks and brought floods to Mediaș and Sighișoara [43] (p. 146). "(On) 1 April, Easter was with snow and strong wind (and) cold until the Sunday before Pentecost, and there was no sweet rain, but always snow mixed with cold winds. ( . . . ) from Easter to the day (Ascension Day; 10 May) there is always snow and cold wind. 26 July, ( . . . ) it started to rain until 15 August, which resulted in a great flood of water, which you have never seen or heard before in the memory of men, with very great damage to fruit, hay, and grass. Many buildings, houses, and mills are torn in several places, and the rainy weather lasted until 15 August. Also, in Burzenland (Țara Bârsei), 6 people died, and (sic!) drowned." [44] "Anno 1668, October. After long rainy weather, there is a very nice, dry autumn without rain for a long time. However, quite a bit of snow fell early on 20 October. And this year, the wine turned out well in Wallachia and there were all sorts of fruits in abundance". In contrast to the two previous years, drought dominated in 1669. Extreme heat and a wasp infestation prevailed in summer [35] (p. 172). "Anno 1669. From 6 August to 6 November, there was no rain in Transylvania; after that, snow fell very early." [44] (p. 19).

The year 1670 was another rainy year; especially in March, April and June it rained continuously, and on 13 August a part of the city walls of Brasov collapsed due to flooding. In Moldova it is said to have rained constantly day and night in early summer. Fruit and wine growing suffered from inundations in summer and frosts in autumn [45] (p. 187). Floods are also reported from the following year 1671 [19] (p. 123). 1672, however, was another year of drought; there was price increase and famine [32] (p. 48). "Hoc anno is such a drought that you could not grind, but had to use the mortar and handmills." [46] (p. 345). In 1673, people groaned under a hot summer, which also brought hail; but there was "a rich harvest of grain and wine". [32] (p. 48). In 1674 there was a cold but dry winter; Water shortage everywhere, the mills stopped turning and there was an increase in prices [32] (p. 48).

In 1675, the winter was severe and the summer was wet. Landslides and water damage have been reported from everywhere and the grapes did not ripen [32] (p. 48). In 1676 a hot summer was recorded [19] (p. 123). In 1677 the winter was hard and the summer was rainy [35] (p. 196, 352). 1678 had a very hot summer [19] (p. 123) and the Autumn was hot and dry too. "At this time of autumn it hadn't rained for several months and it was very dry weather, so that all you could do was sow in the dust; in the forests (with almost no exception) there were a lot of fires everywhere, which could not be extinguished, and so the most delicate forests burned with the many piles of wood in them, which resulted in no small damage." [45] (p. 199). 1679 was a humid year [19] (p. 123).

In 1680, as well as two years before, there was a hot summer [19] (p. 123), and a severe winter struck 1681 [19] (p. 123). 1682 was a year of famine. Furthermore, floods hit the country of Transylvania [19] (p. 123). A thunderstorm in late winter was reported in Țara Bârsei: "On March 18 there was such a thunder and lightning as they are usually seen in the very highest of summer. This storm also hit a Vlach [Romanian] in Zernest [Zărnești] together with his wife so that they both died. Afterward, a lot of snow fell." [45] (p. 202). In 1683, one of the heavy winters was recorded [19] (p. 123), which occurred so frequently at that time. Also, in 1684 the winter was rough [47] (p. 122). Even south of the Transylvanian Alps, people complained about a very harsh winter in Wallachia [33] (p. 387). The year was also very rainy. In Moldova, it is said to have rained continuously for four weeks in autumn. Finally, a cockchafer ("May beetle") infestation also occurred this year [35] (p. 380f).

In 1685 there was another harsh winter [19] (p. 123). "The severe cold spoils the trees." [48] (p. 178). The warm season was severely shortened this year. "When people were on the hay-meadows on 12 September, it snowed an eighth of a snow (sic!) in the night." [45] (p. 215). In 1686, the winter was long. On 15 April, it was snowing heavily at Thorenburg (Turda) [49] (p. 503). The summer was very dry [19] (p. 99). 1687 was a rainy year [19] (p. 123), and in late spring, there was a bad cold snap: "May 26 (there) is such a

cold that even sheep freeze to death." [46] (p. 99). In 1688 the winter was rough [19] (p. 123). Rain and hailstones characterize the year 1689 [19] (p. 123). "In July such a hail fell (in Țara Bârsei) that the hailstones were as big as a walnut, as this had never been seen before; that spoiled the crops." [40] (p. 114).

In 1690, locusts invaded Moldova in such numbers that the sun was said not to have been seen for two months [35] (p. 118f). Then they invaded Transylvania and darkened the sun over today's Cluj-Napoca. The summer was very hot [43] (p. 327). "After a year of drought, there was again a terrible famine and price increase in Biertan." [50] (p. 138). In 1691, on 24 May, a terrible storm struck Sibiu, Brașov, and Codlea, and on 26 June, there was a devastating hailstorm; a little later, locusts invaded the country, and it was very hot and dry [51] (p. 441). "On the day of the Ascension of Christ (24 May) a fierce storm developed and moved over the Carpathian Mountains near Sibiu, where it divided. In Bucharest, it did great damage by striking the royal castle and lighting the powder in a tower, smashing nearby buildings. Still, the other (thunderstorm) approached our Burzenland (Țara Bârsei) and threw a thick hail to the fields, which smashed all the grain in the field of Zeiden (Codlea) down to the roots, killing even crows, sparrows, and other small birds so violently that all the furrows and fields were full of skeletons; in some places, the earth was covered to the height of half a cubit. But the hailstones resembled a big chicken egg and could not melt in many days." [52] (p. 301). In 1692, there was drought. In August, a large number of locusts moved across the authorities of Hermannstadt (Sibiu) and Mühlbach (Sebeș) [53] (p. 88f). 1693 was also a dry year. A cockchafer plague was followed by whole "clouds" of locusts that moved across the country. The summer was exceptionally dry [47] (p. 130). Inundations are reported from 1694 [19] (p. 123).

The year 1695 was also characterized by floodings [19] (p. 123). In autumn of that year, a cold spell damaged the wine cultures: "5 October, the cold spoils the wine." [54] (p. 408). Calamities have also been reported from 1696: on 14 May hoarfrost [55] (p. 146), later floods [19] (p. 123). "On August 21, great hail fell on the field of Zeiden [Codlea], Brenndorf (Bod), and Heldsdorf (Hălchiu), which injured people and killed many birds and threshed the grain" [42] (p. 115). In 1697, there were inundations [19] (p. 123), while in 1698, the summer was particularly hot [19] (p. 123). In 1699, torrential rain fell over Transylvania between June 2 and 5. Particularly in the administrative district of Sibiu and in the mountains of the Siebenrichtergebirge (Munții Lotrului) there were violent cloudbursts so that the river Cibin inundated the fields and flooded the swamps and ponds outside the city walls of Sibiu; the mountain streams of the so-called Transylvanian Alps south of the city swelled to torrents, which tore down mills and huts. In Sibiu itself, pouring streams flowed through the alleys, and water entered the houses through doors and windows [56] (p. 284).

In 1700, there was a very severe and prolonged winter; only in April did the snow melt [57] (p. 57). The 18th century also began with the grim winter of 1701 [19] (p. 205). In 1702 there was rainy weather all over the country and large, "never heard of" inundations [16] (p. II.28). 1703 was a dry year [19] (p. 205) and even more 1704, where "terrible drought" is mentioned [16] (II, p. 28).

Both 1705 and 1706, however, were described as rainy years [58] (p. 171). From the year 1707, there is no climate-relevant testimony from Transylvania. Still, beyond the border, in Moldova, the air was so filled with flying locusts that they darkened the sunlight like clouds [59] (p. 23). In 1708, the weather was extremely dry [19] (p. 205), and the winter was so mild that it was practically non-existent at least at the beginning of the year: "Anno 1708. Just as we had ended the past year with a lovely summer's day, we started this one with the most pleasant spring weather and brightest sunshine; the oldest people in this city testified, not without surprise, that during their lifetime they had no idea of such May weather in the winter months." [60] (p. 289). There was no rain in Honigberg (Hărman) from August 21 to October 11 [40] (p. 115). As extremely mild as the winter was that year, it showed itself from the grim side in the following of 1709: "It is so cold at the beginning of the year that many people, cattle and trees freeze to death, and the waters freeze to the bottom [48] (p. 178). In June, locusts invaded Transylvania via the Bran pass [33] (p. 395).

"When Charles XII (of Sweden) fled to Bessarabia this year, there were swarms (of locusts) from the Black Sea that rose like a storm wind and fell like hailstorms that his infantry and cavalry could neither see, nor get away." [29] (p. 314).

In 1710, there was a terrible drought accompanied with unbearable summer heat; in Moldova locusts filled the air, darkening the sky and causing a terrible roar. They left the forests bare as if it were winter [35] (p. 317f). When they came to Transylvania, they began to kill each other, so that people waded in their carcasses up to the ankles [61] (p. 441). "Again the strongest springs dried up and also many water-rich rivers in such a way that one had big problems with grinding." [48] (p. 178). The drought also had tragic consequences due to superstition: "(On) August 15, a weather maker from Wolkendorf (Vulcan), had put in the spring the dew from 3 meadows into a sack of leather so that it would not rain for a long time. A prolonged heat followed this in Rosenau (Râșnov), so she was sentenced to death in prison . . . On 21 September, white and red roses bloomed, and in some places, the fruit trees also bloomed like in the spring." [51] (p. 458). 1711 was a year with another plague of locusts, probably because of the weather: In Moldova, people complained about unbearable heat and drought in summer [35] (p. 559ff), while in Transylvania floods in the area of the so-called Drei Stuehle took many mills with them; The damp weather also caused the grapevines to rot, and only little and inferior wine was made [61] (p. 475). In 1712, after inundations in spring, an exceptional drought occurred and led to a poor harvest [62] (p. 371). In autumn, locusts fell in again and hibernated in the Maramures region. In 1713, there was no sign of dry conditions. On the contrary: inundations were reported in summer [16] (p. 57f). In 1714, however, from February 28 to June 20, there was no rainfall in the whole country, causing the grain to spoil [46] (p. 348). 1715, the last year of significantly reduced solar activity, was rainy in Transylvania [62] (p. 319).

It can be seen that quite some years during this period had harsh winters—but also dry summers, which is compatible with a higher frequency of blockings [13], however, this was apparently not a continuous feature of the entire period—at least not in Transylvania (see also Table 2 below). Particularly interesting is the small number of severe winters during the last decades of the MM.

**Table 2.** Synoptic comparison of weather phenomena for Transylvania, Germany, Switzerland and Austria. All entries are discussed in the text and the respective references are given.

| Year | Transylvania | Germany | Switzerland | Austria |
|---|---|---|---|---|
| 1645 | Harsh winter; Locusts; Storms | Normal winter | Long winter | Snowy winter |
| 1646 | Heavy snow on March 29 | Harsh winter | - | Cold winter |
|  | Summer without thunderstorms | Variable Summer | - | Dry summer |
| 1647 | floods | - | - | floods in August |
| 1648 | Harsh winter | Normal winter | - | - |
| 1649 | Long and Harsh winter with much snow cold and wet Fall | Long and Harsh winter cold and wet Fall | Normal winter | Normal winter |
| 1650 | Harsh storm on March 10 | Normal conditions | Normal conditions | Normal conditions |
| 1651 | floods in January | Normal winter | Normal winter | Normal winter |
|  | Big snow and very cold at end of April | Cold April and May | - | - |
| 1652 | Harsh winter | Very mild winter | Normal winter | Normal winter |
| 1653 | - | Normal conditions | Normal conditions | Normal conditions |
| 1654 | Winter lasting until April | Only second half of winter cold | - | - |
|  | New winter arrives in September | New winter arrives in November | - | - |
| 1655 | Harsh, long lasting winter with a lot of snow | Very changeable winter temperatures | - | Harsh winter |
| 1656 | Harsh winter | Harsh winter in the south of Germany | - | Harsh winter |
| 1657 | Harsh winter with interruptions | Mild winter in most parts of Germany | - | Harsh winter |
| 1658 | Harsh winter | Extremely cold and long winter | - | Cold winter with lots of snow |
|  | New winter arrives in October | Cold September, warm November |  |  |

**Table 2.** *Cont.*

| Year | Transylvania | Germany | Switzerland | Austria |
|------|-------------|---------|-------------|---------|
| 1659 | Long and Harsh winter | variable winter | - | Extremely cold winter |
| 1660 | Harsh, but dry winter | Harsh winter | - | - |
|      | Drought | Few rain in summer | - | - |
| 1661 | Harsh, but dry winter | Winter mild and dry | - | - |
|      | Drought | Dry conditions in spring and summer | | |
| 1662 | Harsh, but dry winter | Mild winter | - | Mild winter, few snow in Bohemia |
|      | Drought | - | - | - |
|      | Snowfall on October 10 | - | - | - |
| 1663 | Harsh, but dry winter | Extremely cold and long winter | - | - |
|      | Drought | Summer wet and cold, dry autumn | - | - |
| 1664 | Harsh winter | Winter starting mild, but getting cold | - | - |
|      | Drought interrupted due to heavy summer rains | Normal summer, rainy autumn | - | - |
| 1665 | Very cold, snowy winter | Very cold winter | Very cold winter | Very cold winter |
|      | Summer hot and dry | Hot summer | Hot summer | Hot summer |
| 1666 | Harsh, long lasting winter with a lot of snow | Rather mild winter | - | - |
|      | A year of great fertility | Summer very hot and dry | very hot and dry summer | very hot and dry summer |
| 1667 | Rainy year | - | Snow in June | Rainy year |
|      | Big snowfall on September 12 | Big snowfalls starting mid November | - | - |
| 1668 | Long winter | Winter mild, short cold phase in March | - | - |
|      | Rainy summer | cold and rainy summer | - | - |
|      | Snowfall on October 20 | Warm December | - | - |
| 1669 | Very dry and hot summer | Very dry and hot summer | Very dry and hot summer | - |
| 1670 | Persistent rain from March to June | | - | Rainy summer |
|      | Cold autumn | - | - | - |
| 1671 | floods | Mild conditions during whole year | - | floods in Friuli |
| 1672 | Drought | Rainy during second half of year | - | cool and wet summer |
| 1673 | Hot summer, rich crops | Summer cool and rainy | Heavy rains in early summer | Rainy summer |
| 1674 | Harsh, but dry winter | Harsh winter | - | Wet year in Bohemia |
| 1675 | Harsh winter | Very mild winter in the south of Germany | - | - |
|      | Wet summer | Summer cool and wet | Snow in summer | floods at Praha |
| 1676 | Hot summer | warm and pleasant summer | warm and pleasant summer | Good wine |
| 1677 | Harsh winter | Great regional differences in winter temperatures | - | - |
|      | Rainy summer | Cool summer, wine mediocre | Summer dry, good wine | - |
| 1678 | Very hot summer and autumn | Summer very hot and dry | - | hot, but wet summer |
| 1679 | Wet year | Summer stormy and rainy | - | Wet year |
| 1680 | Very hot summer | July hot and dry | Autumn warm and dry | Wine in big quantity |
| 1681 | Harsh winter | Harsh and long winter | - | - |
| 1682 | floods | Inundations | - | Wet year |
| 1683 | Harsh winter | Harsh winter only in East Prussia | - | Harsh winter |
| 1684 | Very Harsh winter | Very Harsh winter | Very harsh winter | Very Harsh winter |
| 1685 | Harsh winter | Harsh and long winter | Severe and very long winter | - |
| 1686 | Long winter | Mild winter, not long | Short winter | - |
|      | Drought in summer | Very warm summer | - | Good wine |
| 1687 | Great cold on May 26 | Many snowfalls in April | Snow in April | - |
|      | Rainy year | Summer cool and wet | - | Rainy autumn |

**Table 2.** *Cont.*

| Year | Transylvania | Germany | Switzerland | Austria |
|------|--------------|---------|-------------|---------|
| 1688 | Harsh winter | Normal winter | Normal winter | Normal winter |
| 1689 | Strong hail shower in July | Summer warm and dry | - | - |
| 1690 | Very hot summer, drought, locusts | Summer rather wet | - | Wet summer |
| 1691 | Strong storms and hailshowers in May and June | - | - | Strong thunderstorms during summer |
| 1692 | Very hot and dry summer Drought, locusts | Rainy summer Cool and wet summer | - | Summer dry and hot Cool and wet summer |
| 1693 | Drought, extreme in summer; locusts | Hot and dry summer | - | Hot and dry summer; locusts |
| 1694 | floods | Cool and wet summer | - | Wine few and sour |
| 1695 | floods; Cold snap in autumn | Cool and wet summer | - | Very bad wine |
| 1696 | floods; Hail showers | Warm and pleasant summer | - | - |
| 1697 | floods | variable, wet and stormy summer | - | - |
| 1698 | Very hot summer | Cool and wet summer | Snowfalls in August | - |
| 1699 | Heavy rain at beginning of June | Normal conditions | - | Rather bad weather |
| 1700 | Harsh, long lasting winter | Extremely cold winter | - | Extremely cold winter |
| 1701 | Harsh winter | Mild winter | Snow in April | Very few snow in Tyrol |
| 1702 | Rainy year, Great floods | Normal conditions | Normal conditions | Very dry year |
| 1703 | Drought | - | - | Drought |
| 1704 | Drought | - | Pleasant summer | Rainy year in Styria |
| 1705 | Rainy year | - | Summer and autumn hot and dry | Wet year |
| 1706 | Rainy year | - | - | Winter and spring very dry |
| 1707 | - | Normal conditions | Normal conditions | Normal conditions |
| 1708 | Very mild winter | Very mild winter | - | Very mild winter |
| 1709 | Very cold winter | Very cold winter | Very cold winter | Very cold winter |
| 1710 | Very hot summer with extreme drought and locusts | Extreme drought in autumn | - | - |
| 1711 | Wet summer | Wet year | Wet year | Wet year |
| 1712 | Drought in summer | Normal conditions | Normal conditions | Rainy summer |
| 1713 | floods in summer | - | - | Floods in summer |
| 1714 | No rain between February 28 und June 20 | Rainy spring | - | Wet spring |
| 1715 | Rainy year | - | - | Rainy summer |

*3.2. Annual Comparison with Data from Germany, Austria and Switzerland*

Since we focused on sources in German language, it seemed tempting to compare the data from the German settlement area within Transylvania with Austria, Germany and Switzerland. In the following, those geographical regions are understood in terms of their current borders, unless otherwise noted (the results are summarized in Table 2 at the end of this section).

While the winter of 1645 was described as exceptionally cold in Transylvania, Germany had average conditions [15] (p. 150). However, in Styria, in Austria, there was a freezing winter [31] (p. 321), and in Switzerland, the lakes were mostly frozen in January and, there was snow in the lowlands [26] (p. 104). The winter of 1646, was described in Germany and Austria [30] (p. I.515) severe like the one in Transylvania, while the dryness that was rather hinted than complained in Transylvania was lamented throughout Europe [29] (p. 175). 1647 was a rather inconspicuous year in Germany. The inundations that occurred in Transylvania found their counterpart in Vienna, when the Danube flooded its banks in August after days of rainy weather [29] (p. 150). 1648, the year in which the terrible Thirty Years' War ended with the Peace of Westphalia, brought an average winter to Germany [25] (p. 151), so the severity of the Transylvanian winter this year may be somewhat overstated due to the lack of quantification. The severeness of the Transylvanian winter of 1649, however, is supported by reports from Germany, where it also lasted until April and where wolves invaded the villages. And, just as in Transylvania, autumn was described as cold and wet [25] (p. 152). In Switzerland and Austria, however, no exceptional conditions were noted.

The year 1650 was an average year in Germany, with rather mild winter conditions [25] (p. 153). Even in Switzerland and Austria [29] (p. 150), the year was not climatically noticeable, which is in good agreement with the reports from Transylvania. In 1651, there were no unusually cold winter conditions in Germany [25] (p. 153), Austria [29] (p. 252), and Switzerland. From late spring to mid-May, cold weather was also recorded in Germany, accompanied by rain, hail, and even snowfall [25] (p. 154), which corresponds to the above-quoted report by Nekesch-Schuller from Transylvania. The severe Transylvanian winter of 1652 cannot be confirmed in Germany, Switzerland, and Austria. On the contrary: in Germany, the winter was exceptionally mild that year [25] (p. 154). The year 1653, missing from our collection of Transylvanian sources, has not been noticed as unusual in Germany, Austria, and Switzerland either, which may explain the silence of the Transylvanian chroniclers about the weather this year. The year 1654 was either less cold in Germany than in Transylvania, or the chroniclers forgot to mention a few warm periods. Glaser [25] (p. 154) stated that the first half of winter was characterized by temperatures well above average. The river Main did not freeze until mid-February, and the winter cold continued through March. The following winter started in Germany at the end of November, two months later than in Transylvania.

The harsh winter of 1655 is confirmed by Austrian sources [30] (p. I.535), [29] (p. 96), while Germany shows the differentiated picture of a changeable winter with alternating warm and cold periods [25] (p. 155). The situation is similar within the subsequent winter of 1656. Austria also reports it as strict [30] (p. I.537), while Glaser for Germany has very different regional conditions. Only in Southern Germany, according to the sources, does the winter seem to have been severe [25] (p. 155). Even in 1657, winter conditions were quite different regionally. The winter in Germany was generally mild and rainy, but not in East Prussia, where a severe winter is reported. [25] (p. 156) Winter was also harsh in Austria, at least in the province of Styria [30] (p. I.537). 1658 had an extreme winter, and in large parts, especially in Germany, it was long and snowy. The cold was so great that this year's winter is considered to have been one of the worst. The Baltic Sea was frozen, wolves, and other wild animals invaded villages and towns. Even the great rivers and streams froze over. In this respect, it may come as a surprise that the Transylvanian winter was described as harsh, but not as outstanding compared to other contemporary winters or the following one in 1659, which was not particularly severe in Germany, but a lot more in Austria. Peinlich stated, "It was even colder (than the last) winter. The wine was freezing, that you had to split it with axes." [30] (p. I.537).

In Germany [25] (p. 158) and Italy [29] (p. 96), the year 1660 was counted among the harsh winters, and the so-called "Seegfrörne" (freezing of the lake) occurred at Lake Constance, meaning that it froze on the surface. As far as the drought of the year is concerned, there was no complaint about it in Central Europe, Glaser (2008) notes that the summer was characterized by good weather and little rainfall. 1661 was also a dry year in Transylvania, but in Germany this was only valid for spring and the first half of summer. In winter, the conditions were felt to be exactly the opposite. While it was portrayed as strict and dry in Transylvania, Glaser considers the winter in Germany as very mild and humid. Even in 1662, the winter in Germany and Austria [30] (p. I.541) was remarkably mild, while in Transylvania, it was counted as a harsh one. The lack of snow reported from Transylvania was also reported from Bohemia [30] (p. I.320). In 1663, the winter in Germany was characterized by extremely low temperatures and an abundance of snow [25] (p. 158). The drought of the Transylvanian summer found no counterpart in the German countries, on the contrary: June was rainy, stormy and cold there. At the beginning of July, there was warm summer weather for a few days, after which rainy and cool conditions set in. It was only in autumn that it became warm and dry. Additionally, in 1664 conditions were experienced rather differently in Germany than in Transylvania. Only the cold of the second half of winter was also felt in German countries. The year began mild in Germany, but severe winter cold set in in mid-January. Spring and summer were not uncommon, but autumn was characterized by cool and rainy weather [25] (p. 159).

In 1665, the correspondence of the conditions in Germany, Austria, and Transylvania is evident. A freezing winter and a hot summer were noted in all three regions. [25] (p. 160), [29] (p. 96). However, the year 1666, was, compared to Transylvania, completely different in Germany, Austria, and Switzerland, where the striking feature was a summer characterized by extraordinary heat and extreme drought [25] (p. 160), [29] (p. 175, Pfister, 131). There is no mention of a harsh winter there—on the contrary: Germany experienced an ordinary winter with a rather mild character [25] (p. 160). 1667 was also a rainy year in Austria [29] (p. 151). Summer cold was reported in Switzerland, and in mid-June, there was knee-deep snow in the hilly country of Central Switzerland [26] (p. 144). The onset of winter in Germany was later than in Transylvania, but still early: In the middle of November, suddenly heavy snowfall set in [25] (p. 160). 1668 was a humid and mild winter in Germany. In March, there was a brief cold spell. The summer was cool and so rainy that the harvest had to be postponed to September. December was so mild in the east of the German Empire that the flowers came out. At Ulm, however, the Danube froze at the turn of the year [25] (p. 161). Overall, the conditions were not unlike those in Transylvania. The exceptional drought and summer heat of 1669 was confirmed in Germany [25] (p. 161) and Switzerland [26] (p. 131). Despite inundations in Tyrol in July, the Austrian summer seems to have been similar; Pilgram particularly highlights the excellent wine of this year [29] (p. 252).

The summer of 1670 was also rainy in Austria [30] (p. II.440), but in Germany it was pleasantly warm [25] (p. 161). While the chroniclers in the Principality of Transylvania did not complain about winter cold this time, a hard winter was reported from Germany [25] (p. 161) and Austria [30] (II p. 440). The inundations that hit Transylvania in 1671 found their counterparts in Friuli [29] (p. 151). In Germany this year attracted attention due to its mild weather all year round [25] (p. 162). 1672 was in Germany and Austria completely contrary to the conditions in Transylvania. From mid-July, rainy weather appeared in Germany and more or less characterized the rest of the year [25] (p. 162). It was a cold and wet summer in Austria [30] (p. I.552). The situation was similar in the following year 1673, when the summer was dry in Transylvania but wet in Austria, Switzerland, and Germany, where the summer was also cool [25] (p. 162). In Switzerland, rains at the beginning of summer led to inundations [26] (p. 161). As a result of constant rains in Austria, the Inn River flooded its banks, followed by the Danube in July, while in Germany the Elbe inundated in August [29] (p. 151). In 1674, winter was hard in Germany. The three winter months were frigid, and the sea froze on the coasts [25] (p. 163). In Western Central Europe, there was no trace of the drought prevailing in Transylvania, on the contrary, it was a very wet year in particular in Bohemia [30] (p. I.552) and the wine yields were few and poor [29] (p. 252).

In June 1675, snow lay in Bern. In July, people went out with thick jackets, and in August, snow fell in the village of Einsiedeln at 880 m above sea level. Higher alpine pastures remained buried under the snow cover all summer [26] (p. 156). In Southern Germany, winter was abnormally warm. Flowers are said to have grown in mid-January. Spring was rainy, and cool, wet weather and inundations prevailed in summer, so the wine was bad and sour [25] (p. 164). The summer of 1676 brought beautiful, warm weather to large parts of Germany [25] (p. 164) and Switzerland [26] (p. 156). The vintage quality was praised in Austria [29] (p. 252), which indicates a hot, dry summer as it was experienced in Transylvania. The winter of 1677 was not perceived as exceptionally cold in Western Central Europe, although it varied widely from region to region in Germany [25] (p. 164). In Switzerland, the late summer was dry and yielded good wine [26] (p. 166), but in Germany, the summer was cool so that the wine was only mediocre [25] (p. 164). 1678 was also extremely hot and dry in Germany. The grape harvest started early, and the wine tasted excellent [25] (p. 165). In Austria, the summer was hot, but not dry [30] (p. I.130). The wet annual characteristic of 1679 in Transylvania matches the reports from western Central Europe. According to Glaser [25] (p. 165), the summer in Germany was stormy and rainy, and [29] (p. 151) notes for Austria: "The famous preacher in his day; Abraham [a Sancta Clara], an eyewitness, complains about the unstable weather and frequent rain,

which arose from the great plague that broke out here in mid-July. So the spring, and (at least) the first part of summer was damp".

In 1680, in Germany, people complained about the heat and drought starting in July [25] (p. 165). A very dry autumn, in which the temperatures were so high that the pear trees could be seen blooming again, was recorded in Switzerland [26] (p. 166). Quantitatively good vintage is reported in Austria. [29] (p. 252) In 1681, winter was extraordinarily cold in Germany, as was the spring. Increasing drought and heat, which then became extreme in summer, plagued Germany and Austria [30] (p. II.131) and Switzerland [26] (p. 166). In the mild winter of 1682, inundations occurred in Germany [25] (p. 166). Austria experienced "unfortunate weather, its main character was wetness, even in summer; pestilence." [30] (p. II.131). In 1683, winter in Germany was only of average cold, or even too mild regionally; only in East Prussia was it clearly severe [25] (p. 166). Lake Traunsee in Austria was frozen [28] and even the water near the coasts of the Adriatic is said to have frozen. [29] (p. 97). In England, this winter was so cold that the Thames got 11 inches of ice [30] (p. II.460). The subsequent winter of 1684 was extreme in all German-speaking countries. In Germany, it was "cruelly cold" and fairs were held on frozen rivers. At the end of March, there was ice on the streets in Königsberg (Kaliningrad) [25] (p. 168). In Switzerland, wine turned to ice in January and February, and trees split with a loud bang [26] (p. 92). Heavily loaded wagons moved across the frozen Rhine, and Lake Constance could be traversed from Hagenau to Münsterlingen on foot [26] (p. 93). Additionally, in Austria, the cold was noted as "extraordinary" [30] (p. II.137).

In 1685, the reports from Germany and Switzerland confirm the news from Transylvania again. Only in Austria, the winter cold seemed not worth mentioning to the chroniclers. In Germany, the winter was very severe, and Lake Constance froze as in the previous year. The sea at the coasts was covered with ice, and it was snowing on many days. Storms and snowfalls lasted into April, and there was still ice on the streets of East Prussia until the middle of the month [25] (p. 168). The cold winter of this year did not seem to end for people in Switzerland either. Rivers and lakes were covered in thick ice. Even in the middle of the summer, on July 26th and 27th, it was still snowing at lower altitudes, even in Zurich (400 m above sea level), snow is said to have fallen [26] (p. 102). In 1686, winter was only in Transylvania considered to be very long. In Germany, the winter was also very mild [25] (p. 169), and from Switzerland, one even finds the opposite of what the Transylvanian chronicler reports. Vegetation started very early, at least in the eastern Alpine valleys. Austria only commemorates a good wine year [29] (p. 253), which agrees well with the Transylvanian summer drought. he summer was also hot in Germany [25] (p. 169). The unusual cold that hit Transylvania at the end of May 1687 had its counterparts in Germany and Switzerland. In the first part of April, it snowed almost every day in Leipzig, but it suddenly became very hot in May. In summer, cool and rainy weather dominated, as in Transylvania [25] (p. 169). In Switzerland, there was snow throughout April in the village of Einsiedeln at 880 m [26] (p. 113). From Austria it reads "excellent summer with wine and fruit in such abundance that the branches broke under their load; afterward, rainy." [30] (p. II.143). The cold Transylvanian winter of 1688 did not have a counterpart in western Central Europe. There was an average winter in Germany [25] (p. 170), and it was not considered to be unusual in Austria and Switzerland either. In 1689 the spring was cold and wet in Germany, but summer was warm and dry [25] (p. 170). In Switzerland and Austria, the year was primarily characterized by many avalanches in winter [26,30] (p. II.461).

"Huge locust swarms in Hungary" [30] (p. II.145), are mentioned in 1690 on the occasion of the description of that year's wet summer in Austria. The year was also marked by moisture in Germany, although the summer varied from region to region [25] (p. 170). The hot and dry summer in Transylvania and the generally wet weather conditions in Western Central Europe may have been responsible for the fact that locust swarms accumulated in extreme density in the eastern parts of Central Europe. While the summer of 1691 was rainy in Germany [25] (p. 171), it was rather dry and hot in Austria. Throughout the summer,

there were violent thunderstorms without cooling the air [30] (p. II.147). The conditions were not unlike those in Transylvania, but locusts again spared Europe's western parts. In 1692 they also haunted Transylvania but spared the Holy Roman Empire, where the summer was generally rainy and chilly [25] (p. 172), [29] (p. 152). In the following year, 1693, summer drought and heat also prevailed in western Central Europe [25] (p. 172); [27] (p. 114), so that locusts penetrated Austria [31]. In 1694, Germany experienced a cool and rainy summer [25] (p. 173). From Austria, it is mentioned that the wine was not only sour but also very little [29] (p. 253). In Tyrol, the year was wet and cold. In the Bolzano area, however, summer was said to be hot and dry [27] (p. 162).

The summer of 1695 was cool and rainy in Germany [25] (p. 173). Pilgram [29] (p. 253) notes "Miserable wine" for Austria. The conditions were similar to the Transylvanian. Poor conditions characterized the whole of June in 1696, but summer became warm and pleasant, at least in Southern Germany [25] (p. 174). 1697 stood out in Western Europe due to an extremely cold and snowy winter [25] (p. 174) [29] (p. 97), [30] (p. II.151). Spring was normal in Germany, summer unstable, and wet and stormy weather prevailed at harvest time [25] (p. 174). In 1698, conditions seem to have been contrary between the Holy Roman Empire and Transylvania. A long, heavy, and snowy winter was reported from Germany and Switzerland. Wet and cold weather dominated the rest of the year, and the wine tasted terrible [25] (p. 175). In 1699, Transylvania was incorporated into the Habsburg Empire, but in terms of climate, the year was unremarkable.

Not only in Transylvania was the winter of 1700 extreme, in Austria, people and livestock froze to death, and even the fish in the ponds perished. "On March 15, it was still so cold that the spittle turned to ice before it fell to the ground." [28] (p. 101). In Tyrol the temperatures likely dropped to below $-30\ °C$, since some church bells cracked. In February there was also a lot of snow [27] (p. 265). It did not look any better in the north of the continent: "The Baltic Sea was covered with ice 10 miles from the coasts." [28] (p. 101). A harsh winter did not open the new century in Central Europe like it did in Transylvania. In Germany, it frequently rained in the winter of 1701 [29] (p. 207), while in Austria, high temperatures and exceptional drought dominated the year [28] (p. 101). Only in Switzerland, especially at higher altitudes, was there a lot of snow in April [26] (p. 114). The rainy year 1702 had its equivalent in Italy, where it rained almost continuously for four months, beginning at the end of February. On the other hand, great drought was reported from Austria as in the previous year [28] (p. 102). Neither exceptionally dry nor exceptionally moist conditions were reported from Germany and Switzerland. In 1703, the wine became excellent in Austria "despite" an extreme drought [28] (p. 103). While the dry conditions in Transylvania continued in 1704, and Italy also suffered from July to October, summer in Switzerland was regarded as pleasant with sunny weather and not excessive warmth. In Austria, some witnesses considered the year to be even a rainy one, especially in Styria [29] (p. 153).

The year 1705 was rainy in Transylvania, and Austria also had wet conditions most of the year [29] (p. 153). In Switzerland, conditions have been more differentiated: July was hot, and August was also dry with pronounced summer heat, and there was very little snow in the high mountains. In October, Foehn storms swept across the Alps [26] (p. 169). 1706 was also a humid year in Transylvania. In Austria, however, at least winter and spring were very dry [28] (p. 104). 1707 seems not to have been exceptional in Central Europe, while the strikingly mild Transylvanian winter of 1708 also prevailed in Germany [25] (p. 181) and Austria. However, there also occurred heavy snowfall there regionally [28] (p. 105). Richard Peinlich clarifies: "It was so mild in the winter that the flowers bloomed around Ödenburg (Sopron) in February, but one day the cold in Styria was so strong that the wine froze in Pettau (Ptui)." [30] (p. II.156). The severeness of the winter of 1709 prevailed in large parts of Europe, where it became known as "great frost" or "great winter". Glaser mentioned an extremely harsh winter for Germany [25] (p. 181), Peinlich in Austria called it the "coldest winter in a hundred years; the length and roughness of the same is said to be extraordinary everywhere; snow still fell in Ödenburg on May 17". [30] (p. II.157). In

Switzerland, the cold in January was so great that the cattle were taken into the parlors in some places to save them from freezing to death. Birds are said to have fainted on the ground so that they could be caught by hand [26] (p. 102).

The year 1710 brought extreme drought to Germany, but only in autumn [25] (p. 182). In contrast, the following year, 1711, it was slightly humid in Western Central Europe [29] (p. 153; Strömmer, 2003; 108). 1712 was unremarkable in Germany and Switzerland. In Austria, the harvest failed as in Transylvania, but for a different reason. There were frequent storms in summer. The wine became delicious but was available only in very reduced quantities [28] (p. 108). "Very changeable weather, so the harvest is bad", is reported from the province of Styria [30] (p. II.178). The summer floods in Transylvania in 1713 had their equivalent in Austria. After a rainy May, there were inundations in June, which destroyed mills [28] (p. 109). The dry conditions of 1714 did not show up in Western Central Europe. The weather in Germany, Austria, and Switzerland was very unstable and precipitation was rare [25] (p. 183). In 1715, the conditions were finally rainy, not only in Transylvania but also in Austria, at least in summer. Winter was temperate and dry in the province of Styria, but after that, the weather was even more changeable and unhealthy than last year's [30] (p. II.230).

### 3.3. The Decades before and after—Comparison with the Fluctuations in Solar Activity

As part of a research project for the Austrian Science Fund (FWF project P 31088), we have created a still unpublished database that compiles climate-relevant sources in German language and traces the climate of Transylvania back to the year 1500 CE in almost annual resolution [11]. From this data set, we counted the reported unusually cold or mild winters and particularly hot or cool summers per decade and compared these numbers with a graph of the reconstructed strength of the solar activity (Figures 2–4), as published by Wu et al. [63] which ends in 1889. For the later decades up to 1950, we used the annual SILSO sunspot numbers (Sunspot Index and Long-term Solar Observations, source: Source: Royal Observatory of Belgium, Brussels [64]). While decades are usually counted starting with the year "1", there is an alternative counting that defines decades from the 0 years to the years ending with 9. Since the reconstruction by Wu et al. [63] uses the latter, we also adopted this scheme for better compatibility. Although International sunspot numbers (ISN) are listed as far back as to the year 1700 CE within the SILSO dataset, we decided to use the reconstruction of Wu et al. [63] instead for two reasons: Firstly we wanted to avoid as far as possible inhomogeneity within the series (which would have occurred within the MM period by switching from one data set to an-other) and secondly, the early SILSO data are currently under revision by WDC-SILSO, so they are not used here until 1890.

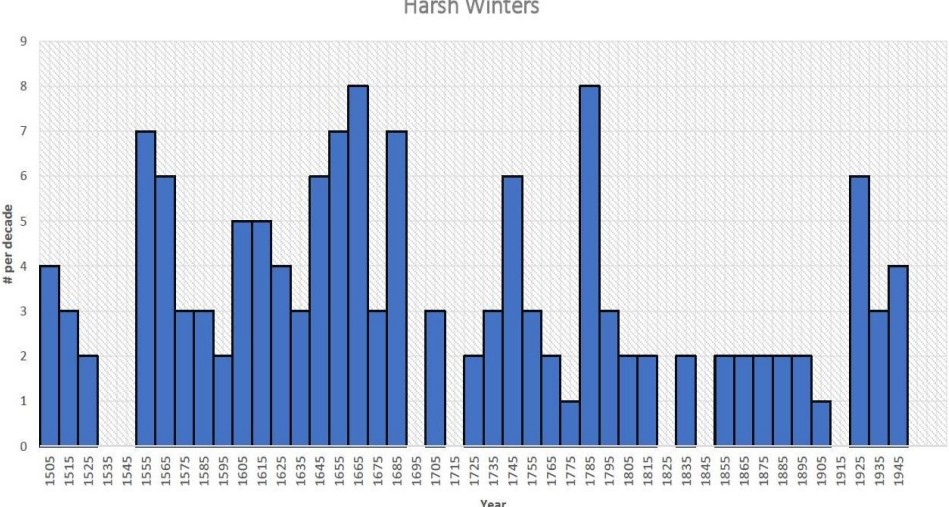

**Figure 2.** *Cont.*

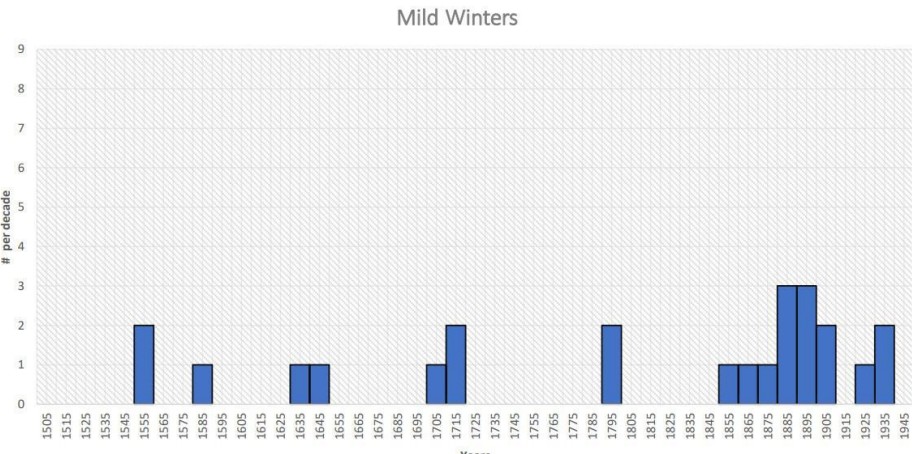

**Figure 2.** Decadal numbers of harsh winters (**top**) and mild winters (**bottom**) in Transylvania for the period of 1500 to 1950.

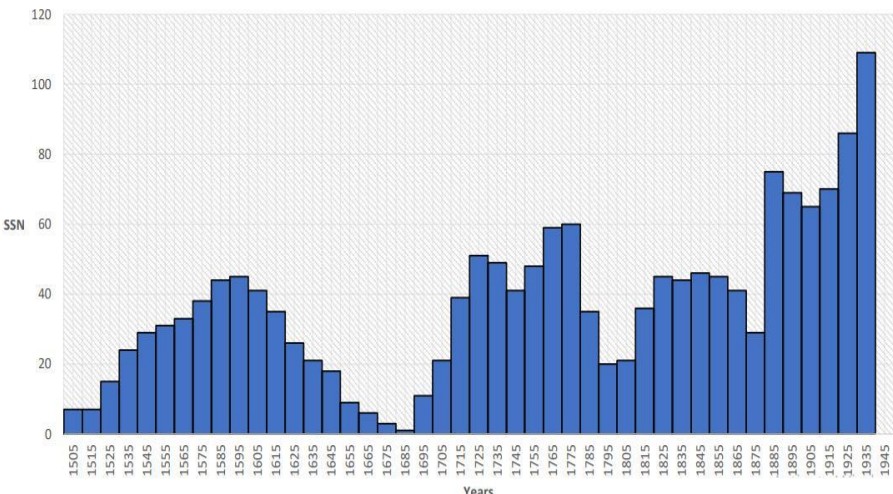

**Figure 3.** Solar activity in the years of 1500 to 1950, given in decadal sunspot numbers (SNN). The time period of 1500 to 1890 is based on the reconstruction by Wu et al. [63], while actual sunspot numbers (directly observed or photographed) were used for the later decades (SILSO data, Royal Observatory of Belgium, Brussels).

Two large and two small minima of solar activity fall within the investigation period. The two large ones are the Spörer Minimum (1450–1540) and the Maunder Minimum (1645–1715), the two small ones are known as the Dalton Minimum (1790–1830) and the Gleissberg Minimum (1901–1912).

It is commonly believed that in years of high solar activity, temperatures on Earth will rise, and mild winters and hot summers will increase in frequency. In contrast, the number of severe winters and cool summers is expected to increase with decreasing solar activity. For our study region, the database is not sufficient for a thorough statistical analysis. However, we can at least compare the two time series to see if there is any indication for a possible relation between solar activity and climatic conditions.

When comparing the top panel of Figure 2 (number of hard winters) with Figure 3 (solar activity), an accumulation of severe winters appears in the time of the abating Spörer minimum (note that this minimum is only partly covered by our data) and also over long periods of the MM. In contrast, for the less pronounced Dalton and Gleissberg minima, no accumulation of cold winters is noticeable. However, it should be noted that there is no convincing correlation, since the number of exceptionally cold winters has also high peaks in the middle and at the end of the eighteenth century and even after 1920 when the solar activity was high to very high.

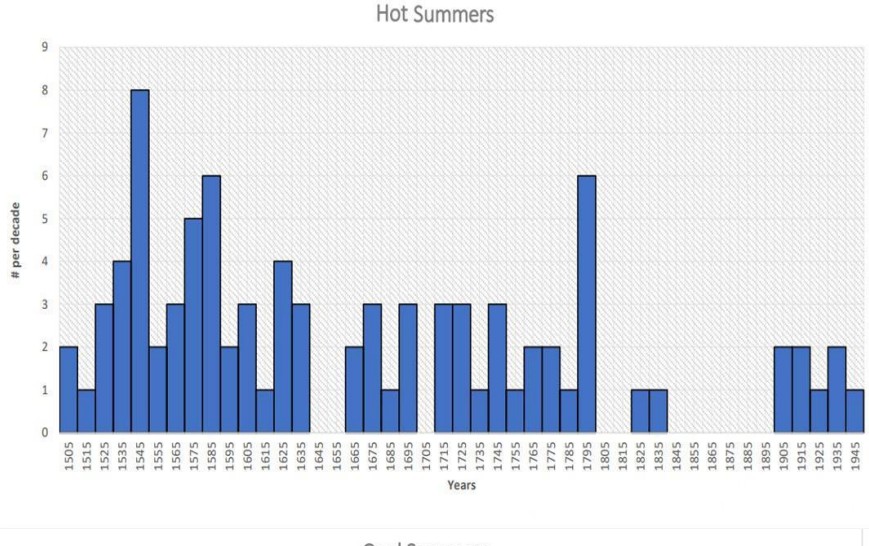

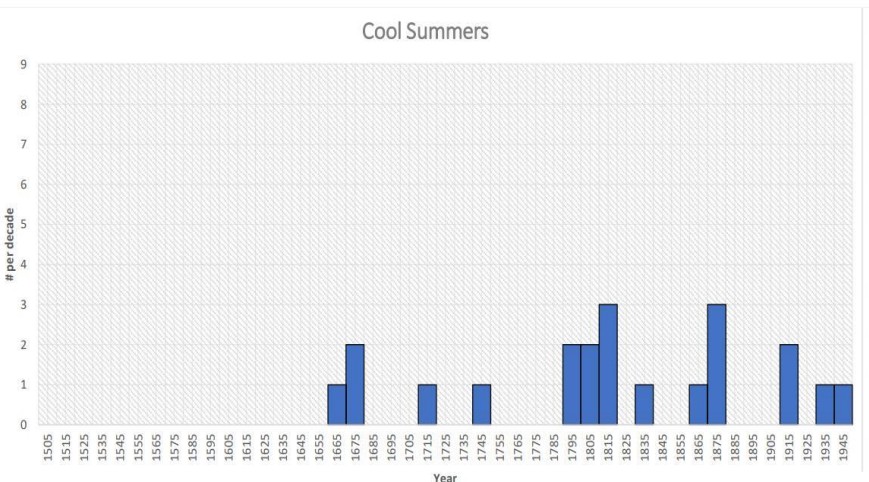

**Figure 4.** Decadal numbers of cool summers (**top**) and hot summers (**bottom**) in Transylvania for the period of 1500 to 1950.

Let us consider the bottom panel of Figure 2, which shows those winters that the contemporaries considered exceptionally mild. A "negative picture" of the top panel of Figure 2 would be expected this time: high numbers at times of high solar activity. The (moderate) peaks around 1555, 1585 and 1635–1645 somewhat match the moderate to strong solar activity between the two grand Minima Spörer and Maunder between 1525 and 1645, as do the peaks from 1705–1715 with the renewal of solar activity at the end of the MM. However, the peak around 1795 would not have been expected because the sun's activity had subsided again due to the shallow Dalton minimum, while the peaks from 1885 still fit well with the high solar activity of that time.

The top panel of Figure 4 shows the cool summers of the examination period. Their accumulation should again go hand in hand with low solar activity. In fact, the peaks from 1665–1675 fit well the most pronounced phase of the MM. Otherwise, the remaining peaks appear without exception at times of high solar activity.

The bottom panel of Figure 4 finally shows the hot summers, the peaks of which should correspond to the peaks of solar activity. While for the time between the Spörer and the Maunder Minimum the agreement is quite good and likewise for the time between Maunder and Dalton Minimum, it must be noted that distinct peaks fall in the very middle of the MM, where none were to be expected.

## 4. Discussion

It should be noted that historical sources typically characterize entire seasons with just a few sentences, while actual measurements provide a more detailed view. The winter of 1708/09, as an example, is usually described (in many parts of Europe) as the coldest winter of the century (or similar). Actual temperature measurements have just been taken at few locations. In a different study [65], we analyzed sub-daily temperature measurements by Louis Morin in Paris, which deliver a more differentiated view of this extreme winter. In Paris, the extreme cold was restricted to about three weeks in January and a short period in late February—while the rest of the winter was quite moderate. Historical sources will seldom provide such a detailed view; however, they are important since actual measurement time series are very limited.

Although the data presented here for the Transylvanian climate suggest a certain degree of influence from long-term fluctuations in solar activity, one must remain extremely cautious in their interpretation. At best, one could see this as an indication and wait for future evaluation of climate data from the entire German-speaking area. From a European perspective, Transylvania is only a small province, and the data are much sparser than for Germany, Austria, or Switzerland. We would like to emphasize that previous studies in this area, which had data on a much larger, and geographically broader basis, have not led to any convincing results regarding solar forcing [6]. The sometimes seemingly striking agreement in parts of the curves presented here may, therefore, be purely coincidental.

## 5. Conclusions

For our work, we selected the period of the Maunder Minimum (MM, 1645–1715), an astrophysically defined section of the Little Ice Age (LIA), and compared the historical data from the Grand Duchy of Transylvania with those from Germany, Austria and Switzerland. For the first time, we present here a comparison between the four geographic areas in text and tabular form. Quotes from mostly German-language sources are reproduced in English translation. There are numerous similarities, but also geographic differences, so that a convincing picture emerges of the weather and climate conditions prevailing in a region that until 1699 was somewhat outside and then just within the borders of the Habsburg Empire.

Furthermore, for a larger period (1500–1950), we examined on a decadal basis the extent to which an influence on the climate of Transylvania through long-term fluctuations in solar activity, as was inferred from isotope reconstructions from ice cores, can be seen. With the exception of the years 1820–1830 and 1840–1860, there are sufficient historical data to be able to count the frequencies of hot and cold anomalies in the respective summers and winters reliably, so that a picture emerges which decades were characterized by warmth above or below average in the respective seasons. Although this comparison shows some promising result, which suggests a certain solar influence, further studies will be needed to substantiate these findings. In future work, we plan to increase the data density by including other databases, especially ones that rely on sources in Hungarian language, and by including the results from dendrochronological and other proxy studies [63].

Already now, the results clearly help to identify regional climatic differences during the MM. Some results are unexpected—like an unusually small number of severe winters during the last decades of the MM, where extreme cold was restricted to a few years, like the extreme winters 1699/1700 and 1708/1709.

**Author Contributions:** Conceptualization, methodology, validation, formal analysis, investigation, resources, data curation, writing—original draft preparation, writing—review and editing, visualization, M.S. and U.F.; supervision, project administration, funding acquisition, U.F. All authors have read and agreed to the published version of the manuscript.

**Funding:** This paper is part of the project "Climate History of Central Europe during the Little Ice Age", funded by the Austrian Science Fund (FWF), project number P 31088.

**Institutional Review Board Statement:** Not applicable.

**Informed Consent Statement:** Not applicable.

**Data Availability Statement:** Data available in a publicly accessible repository. The data presented in this study are openly available at: https://doi.org/10.1051/0004-6361/201731892 [63] (accessed on 15 November 2021).

**Acknowledgments:** We thank Joachim Wittstock, writer and historian, Sibiu, Romania, for help in identifying potential historical source texts, Andreea Puchianu for translating Romanian sources and Rosalinda Szemcsuk for translating Hungarian sources.

**Conflicts of Interest:** The authors declare no conflict of interest.

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
