# Peer review of "Climate History of the Principality of Transylvania during the Maunder Minimum (MM) Years (1645–1715 CE) Reconstructed from German Language Sources"

_climate, doi:10.3390/cli10050066_

Round 1

Reviewer 1 Report

The authors of the article studied the historical chronicles of the Transylvanian region. They built a generalized record of extreme meteorological events in Transylvania during the epoch of the Maunder Minimum. An analysis of this series allowed the authors to make a cautious conclusion about a possible connection between extreme weather events and changes in solar activity. The work done, of course, is enormous, and the topic of research is important. That is why I believe that the paper can be published after some revision.

COMMENTS.

  • The authors detail the historical evidence for each extreme weather event they have identified. All entries in Table 2 are discussed in the article. I think too many historical texts are cited. Perhaps it would be better to discuss only a few events as an example of the method of analysis. And the description of other events place in Supplementary materials.

  • The authors estimate possible influence of solar activity on Transylvanian weather by means of a simple visual comparison of the corresponding figures. I don’t think that it is enough for any conclusion. At least the coefficient of correlation is desirable. E.g. the authors can smooth the data on harsh winter by 40-50 years, calculate coefficient of correlation with the smoothed by the same way sunspot number and estimate its significance. In any case statistical analysis should be improved.

  • The authors write: Figure 3: “Solar activity in the years 1500 to 1950, given in decadal Sunspot Numbers (SNN). The time period 1500 to 1890 is based on the reconstruction by Wu et al. (2018), while actual sunspot numbers (directly observed or photographed) were used for the later decades (SILSO data, Royal Observatory of Belgium, Brussels).”

I am not sure that Wu et al. (2018),follows to MDPI citation style. Why the authors used the reconstruction of Wu et al. till 1890? The SILSO series is available since 1700 AD. And instrumental data, even based on early telescopic observation, are more reliable than reconstructions, based on proxy data. It is better to use direct sunspot data since 1700.

  • Despite I am not an expert in English, it seems to me that the English can be improved. E.g. I don’t understand well the text: “Although the historical data from Transylvania certainly do have specific gaps and are subject to a certain subjectivity of their reporters, a comparison can be risked after all. Only the decades around 1825, 1845, and 1855 contain such sparse concrete information that the specified climate values are fraught with high uncertaint

Author Response

We thank reviewer 1 for his or her appreciation of our study, including thorough comments and important suggestions to improve the paper.

  • It was indeed our intention to give more room as usual to actual quotes from historical sources. We are aware of the possibility to shorten this section in order to paraphrase the contents, but decided against it since by doing so it gives the reader more insight into the actual sources. That's why we prefer leaving the quotations in the main text and not outsource them into Supplementary materials.
  •  
  • It is absolutely true and we fully agree to the point made by reviewer 1 that no far reaching conclusions should be drawn from the simple correlation found between solar activity and the occurrence of cold/mild winters and hot/cool summers within a specific geographic region as it is represented by Transylvania. Actually from the nature of the data (historical descriptions and proxy reconstructed solar activity), we tried to see if any tendency shows up at all. That is why we did not try any deeper statistical look into the connection in this paper.

Although SILSO data of observed sunspot counts do indeed exist down to the year 1700 CE, we decided to use the reconstruction of Wu et al. (2018) instead for two reasons: Firstly we wanted to avoid as far as possible inhomogeneities within the series (which would have occurred within the MM period by switching from one data set to another) and secondly, the early SILSO data (actually for the whole 18th century and depending to the demands as recent as 1847) are problematic and less reliable than the reconstructions - cf. for example Muñoz-Jaramillo, A. and Vaquero, J. M.: 2018, Visualization of the challenges and limitations of the long-term sunspot number record. Nature Astronomy, 3, 205–211.  

The citation Wu et al. (2018) in Fig. 3 has been corrected to Wu et al. [16]

The sentences " Although the historical data from Transylvania certainly do have specific gaps and are subject to a certain subjectivity of their reporters, a comparison can be risked after all. Only the decades around 1825, 1845, and 1855 contain such sparse concrete information that the specified climate values are fraught with high uncertainty." have been changed to " For our study region, the database is not sufficient for a thorough statistical analysis. However, we can at least compare the two time series to see if there is any indication for a possible relation between solar activity and climatic conditions."

Reviewer 2 Report

The paper introduce of very interesting analysis of climate conditions of Transylvania during the  Maunder Minimum years reconstructed using national historical chronicles. The paper is very interesting especially for scientists working in paleoclimatology. It in scope of J. Climate.

The manuscript can be accepted for publication after some revision.

I guess that the discussion in the paper must be extended. In present version is too short.  I cannot see any conclusion.

Analysis of comparison of climate condition and solar activity (chapter 3.4) is very shallow. I believe further more deep statistical analysis is required.

Specific comments

I didn't find the section 3.3 in the chapter 3.

The axis titles in all figures are small and not good understandable.

The information in table 3 is prepared in graphic format and not of the best quality.

Author Response

We thank reviewer 2 for his or her important observations and benevolent suggestions to improve our paper.

The discussion has been extended and the conclusion section added. See also the response to the "Specific comments" below.

Regarding comparison of climate condition and solar activity (chapter 3.4) we agree that more detailed work must be done in the future in order to cement the proposed connection. However, as in this paper we just tried as a first step to show any correlation at all and rather unexpectedly found good agreement between the historical sources and solar activity data, we feel that for thorough statistical analysis more detailed data as well from the climatic point of view as from the astrophysical one are needed.

Specific comments

There have been several formatting issues, which caused sections missing and/or mislabeled. Section 3.4 in chapter 3 has been correctly renamed now in 3.3 as requested.

The axis titles in all figures have been enlarged.

Table 3 is indeed prepared in graphic format and the letters unfortunately small. We decided to let this unchanged for the moment since we want to check the final form with the editors to see what can be done in order to improve legibility.

Reviewer 3 Report

The paper about Climate history of the principality of Transylvania during the 
Maunder Minimum (MM) years (1645 - 1715 CE) reconstructed 
from German Language Sources has an interesting topic constructed by the use of special sources of information. The paper is well structured and before publishing needs minor revisions regarding language and missing real conclusion.

General comments.

1 in whole the text it can be found a lot of "we" row 20, 36, 87, 95, ...788, 797. We suggest the authors to use the impersonal style in expression.

2. the discussion part is organized like a conclusion part. In this sense the results part can be organized like a results and discussion part.

Author Response

We thank reviewer 3 for his or her observations and recommendations in order to improve our paper.

  1. We replaced "we" wherever it seemed appropriate with impersonal style as suggested, however in some cases we left the formulation unchanged in order not to produce possible misunderstandings.
  2. This was indeed a mistake which has been corrected exactly as requested. There was a mix up within the sections "results", " discussions" and "conclusions", which took place when we reorganized our original manuscript in order to fit the "climate" template. This has now been corrected.

Reviewer 4 Report

A comparative, interesting and proper analysis regarding the reconstitution of climatic conditions, influenced by the solar activity that occured between 1645 and 1715 (MM) for Transylvania, Germany, Austria and Switzerland. It is a good demarche that it's worth continuing and completing with additional data. 

Author Response

We thank reviewer 4 for his or her observations and interest in our paper. We agree that additional data, namely from Germany, Austria and Switzerland, but also from Transylvania (especially sources in Hungarian language) should make for an interesting future study in order to investigate possible solar forcing during the MM in more detail.

Round 2

Reviewer 1 Report

The authors write:

“ Although International sunspot numbers (ISN) are listed as far back as to the year 1700 CE within the SILSO dataset, we decided to use the reconstruction of Wu et al. [66] instead for two reasons: Firstly we wanted to avoid as far as possible inhomogeneity within the series (which would have ocurred within the MM period by switching from one data set to another) and secondly, the early SILSO data (actually for the whole 18th century and depending on the demands as recent as 1847) are problematic and less reliable than the reconstructions [cf.

67]”

As can be seen from Muñoz-Jaramillo and Vaquero (2018), the instrumental sunspot number is quite reliable since 1750. Thus telescopic data are more reliable than cosmogenic proxies since at least 1750. Therefore, I recommend that the authors avoid discussing SILSO and change this section:

“We used Wu et al. [66] until 1890 because we wanted to avoid inhomogeneity within the series as much as possible.”

The authors write: “It is absolutely true and we fully agree to the point made by reviewer 1 that no far reaching conclusions should be drawn from the simple correlation found between solar activity and the occurrence of cold/mild winters and hot/cool summers within a specific geographic region as it is represented by Transylvania”.

The problem is that the authors did not even calculate the linear correlation coefficient, despite the fact that at least in one case (severe winters) it was possible. I recommend that the authors in their future work not limit their analysis only to a visual comparison of data.

Author Response

We seriously considered the suggestion of reviewer 1 to replace proxy generated SSN with SILSO's, therefore we contacted both SILSO itself and one of the authors of Wu et al. in order to shed some more light about which data should be preferred for our study, and received the following answers:

(1)  If one is interested in short-term (i.e. annual) variability, direct SILSO numbers might prove more robust. On the other hand, a long-term drift/trend is likely to exist within the SILSO data which cannot be well calibrated before ca. 1850 and particularly not so during the Dalton minimum (ca. 1790 - 1830). Work on re-calibration of the ISN SILSO series is going on, because long-term variability is somewhat uncertain. Cosmogenic proxies are regarded as very robust in the long-term scale we are interested in our study (cycle-averaged activity), while the exact variability within individual solar cycles is well represented by direct SN counts.

(2)  We have got confirmation from SILSO that data at least until 1816 are not particularly trustworthy and will undergo major revisions within the near future. From 1849 onwards, data have been computed on a daily basis from direct observations by R. Wolf, but from 1818 to 1848, Wolf used, a posteriori, a number of observations from other observers like H. Schwabe, while before that, the observations were collected from unavailable sources. Since Wolf did not make the computation information available, not even the number of days from which they were calculated or the quality of the observations is known. WDC-SILSO, as mentioned above, is working towards a reconstruction from raw data of early Sunspot numbers, so more information will be available for future studies (probably in a few years), but not at the moment.

Taking all this into consideration, we decided, especially with data homogeneity in mind, to keep the graph and section in question as it is, but of course we respect and appreciate the recommendations of reviewer 1 regarding this point and agree, that directly derived SN would be preferable over proxies, but as it is, only after revision of 18th and 19th century data. As direct data before 1849 are still the ones derived almost 200 years ago, and we also don't want to switch between two data sets during "critical" periods including the Dalton Minimum, we prefer to use the Wu et al. reconstruction for the whole time span until 1890.

Following the request of reviewer 1 not to criticize the SILSO data, we changed the sentence "Although International sunspot numbers (ISN) are listed as far back as to the year 1700 CE within the SILSO dataset, we decided to use the reconstruction of Wu et al. [66] instead for two reasons: Firstly we wanted to avoid as far as possible inhomogeneity within the series (which would have occurred within the MM period by switching from one data set to an-other) and secondly, the early SILSO data (actually for the whole 18th century and de-pending on the demands as recent as 1847) are problematic and less reliable than the reconstructions [cf. 67]" into "Although International sunspot numbers (ISN) are listed as far back as to the year 1700 CE within the SILSO dataset, we decided to use the reconstruction of Wu et al. [66] instead for two reasons: Firstly we wanted to avoid as far as possible inhomogeneity within the series (which would have occurred within the MM period by switching from one data set to an-other) and secondly, the early SILSO data are currently under revision by WDC-SILSO, so they are not used here until 1890"  - without citing Muñoz-Jaramillo and Vaquero (2018) anymore.

We also thank reviewer 1 again for drawing to our attention sensible improvement of statistical comparison, which indeed is scheduled for our future work within this domain.